# Calcification of the thoracic aorta on low-dose chest CT predicts severe COVID-19

Philipp Fervers[1]*, Jonathan Kottlors[1], David Zopfs[1], Johannes Bremm[1], David Maintz[1], Orkhan Safarov[2], Stephanie Tritt[2], Nuran Abdullayev[1], Thorsten Persigehl[1]

1 Department of Diagnostic and Interventional Radiology, Faculty of Medicine and University Hospital Cologne, University Cologne, Cologne, Germany, 2 Department of Radiology, Helios Dr. Horst Schmidt Kliniken Wiesbaden, Wiesbaden, Germany

* philipp.fervers@uk-koeln.de

**Data Availability Statement:** All relevant data are within the manuscript and its Supporting Information files.

**Funding:** The authors received no specific funding for this work.

## Abstract

### Background

Cardiovascular comorbidity anticipates poor prognosis of SARS-CoV-2 disease (COVID-19) and correlates with the systemic atherosclerotic transformation of the arterial vessels. The amount of aortic wall calcification (AWC) can be estimated on low-dose chest CT. We suggest quantification of AWC on the low-dose chest CT, which is initially performed for the diagnosis of COVID-19, to screen for patients at risk of severe COVID-19.

### Methods

Seventy consecutive patients (46 in center 1, 24 in center 2) with parallel low-dose chest CT and positive RT-PCR for SARS-CoV-2 were included in our multi-center, multi-vendor study. The outcome was rated moderate (no hospitalization, hospitalization) and severe (ICU, tracheal intubation, death), the latter implying a requirement for intensive care treatment. The amount of AWC was quantified with the CT vendor's software.

### Results

Of 70 included patients, 38 developed a moderate, and 32 a severe COVID-19. The average volume of AWC was significantly higher throughout the subgroup with severe COVID-19, when compared to moderate cases (771.7 mm$^3$ (Q1 = 49.8 mm$^3$, Q3 = 3065.5 mm$^3$) vs. 0 mm$^3$ (Q1 = 0 mm$^3$, Q3 = 57.3 mm$^3$)). Within multivariate regression analysis, including AWC, patient age and sex, as well as a cardiovascular comorbidity score, the volume of AWC was the only significant regressor for severe COVID-19 (p = 0.004). For AWC > 3000 mm$^3$, the logistic regression predicts risk for a severe progression of 0.78. If there are no visually detectable AWC risk for severe progression is 0.13, only.

### Conclusion

AWC seems to be an independent biomarker for the prediction of severe progression and intensive care treatment of COVID-19 already at the time of patient admission to the hospital; verification in a larger multi-center, multi-vendor study is desired.

**Competing interests:** David Maintz has received speaker's honoraria from Philips Healthcare, unrelated to the presented work. David Zopfs has received exemption from clinical duties as part of a research agreement between Philips Healthcare and University Hospital Cologne, unrelated to this project. This does not alter our adherence to PLOS ONE policies on sharing data and materials.

## Introduction

Coronavirus Disease 2019 (COVID-19) caused by severe acute respiratory syndrome corona-virus 2 (SARS-CoV-2) was declared a pandemic by the WHO on 11th of March 2020 [1] and continues to challenge healthcare systems around the globe. While most patients infected with SARS-CoV-2 present only mild and non-specific symptoms, i.e., fever and cough, fatigue, or myalgias [2–5], severe complications include respiratory failure, sepsis and cardiac mortality [6, 7]. More than half of all SARS-CoV-2 transmissions are accounted for by asymptomatic patients, which demonstrates the need for comprehensive testing [8]. A recent comprehensive meta-analysis reported excellent sensitivity (94%) and limited specificity (37%) of LDCT [9]. While the specificity of the RT-PCR test is assumed 100%, sensitivity gradually increases from the day of infection with SARS-CoV-2 (100% false-negative) to a minimum of 20% false-nega-tive at day eight after infection, as another recent meta-analysis showed [10]. An early and accurate detection of COVID-19 infected patients is important for individual and healthcare reasons, since an early estimation of required intensive care capacities is substantial to not overstrain the hospital system. Thus, the Radiological Society of North America suggests diagnostic LDCT for all patients with worsening respiratory status and moderate to severe features consistent with COVID-19 as well as high risk for intensive care treatment in the near future [11, 12].

Besides the evaluation of pulmonary infiltration at the different stages of COVID-19, non-contrast LDCT allows for limited assessment of the thoracic aorta. In particular, calcified atherosclerotic plaques can be detected on LDCT, considering their high intrinsic contrast to the adjacent soft tissue. Vascular calcification and, more specifically, aortic wall calcification (AWC) is the endpoint of an atherosclerotic transformation of the vessel wall [13, 14]. Atherosclerosis reflects the lifetime burden of all known and unknown factors causing systemic atherosclerotic disease [15], including high age, active smoking, diabetes mellitus, dyslipidemia, and arterial hypertension [16]. The magnitude of AWC as an indicator of the burden of atherosclerotic disease correlates with cardiovascular risk factors (CRF) [17, 18] and predicts the likelihood of a cardiovascular event [19, 20]. Therefore, individual CRFs and cardiovascular comorbidities can be estimated by quantification of AWC on a non-contrast LDCT scan.

Based on several clinical papers, severe progression of COVID-19 and poor patient outcome is strongly associated with existing CRFs and cardiovascular comorbidity [3, 21–24]. In two early studies on COVID-19, 63–67% of deceased patients were reported to suffer from cardiovascular comorbidities, most commonly hypertension, diabetes, and coronary heart disease [2, 7]. This suggests that the measurement of AWC can act as a predictor for the COVID-19 outcome and in particular, the requirement of intensive care treatment.

The objective of this study was to evaluate if the magnitude of AWC determined via LDCT is a feasible biomarker to predict the patient's risk for severe COVID-19 progression.

## Materials and methods

All procedures performed in our retrospective study involving human participants were in accordance with the ethical standards of the institutional and national research committee and with the 1964 Helsinki declaration and its later amendments or comparable ethical standards. The study was approved by the ethics committee of the University Cologne by the approval number 20–1367. Written consent of the patients was waived due to prior anonymization of data and retrospective study characteristics. All imaging was performed for clinical indications. No scan was conducted explicitly for the purpose of this study.

## Patient enrollment and follow up

Patients were enrolled from two primary care hospitals in Germany, *blinded* (below center 1) and *blinded* (below center 2). We screened the databases for consecutive patients with LDCT from the 10th of March to the 30th of June, 2020. Indication for the acquisition of LDCT at both centers was defined by clinical considerations. LDCT was performed for patients with

1. moderate and severe clinical features in suspected or RT-PCR positive COVID-19;

2. worsening of respiratory status in suspected or RT-PCR positive COVID-19.

   Inclusion criteria for our study were:

1. a parallel LDCT and mouth swab with positive RT-PCR for SARS-CoV-2;

2. patient age ≥18 years;

3. follow-up of at least 22 days after admission to the hospital-based on reported data [2]. One patient was excluded due to severe beam hardening artifacts, which rendered identification of vascular plaques impossible.

COVID-19 outcome was rated by clinical features on a scale from 1–5 for each patient by a single observer at both institutions, higher numbers representing a more severe progression of the disease (no hospitalization = 1, hospitalization = 2, intensive care unit (ICU) = 3, tracheal intubation = 4, death = 5). Outcomes 1 and 2 were called moderate, while 3–5 were considered severe. Endpoints 3–5 implied obligation for intensive care treatment throughout the included patients. Each patient was observed for 22 days after admission to the hospital. The highest achieved endpoint during the observation period was noted.

## Assessment of clinical risk factors

A basic cardiovascular comorbidity score was investigated for each patient in order to test the robustness of the independent variable AWC against clinical data. The 5-point score introduced below can be determined in a quick interview and by reviewing a patients' daily medication. One point was added for each positive observation from the five following items: 1. history of arterial hypertension or antihypertensive drugs in medication, 2. type 2 diabetes or anti-diabetic medication, 3. prior cardiovascular event (myocardial infarction or revascularization procedure, stroke or transient ischemic attack, diagnosis of peripheral arterial disease), 4. hyperlipidemia or lipid-lowering agents in medication, 5. active smoking habit. The used score respects the three most consistently reported clinical risk factors for severe COVID-19 (cardio-vascular disease, hypertension, diabetes) [25] as well as two additional points for hyperlipidemia and active smoking, which are further assumed risk factors [26–29].

## LDCT scanning protocol and image reconstruction

All clinically indicated CT examinations were performed on CT scanners of two vendors (center 1: iCT 256, Philips and center 2: SOMATOM Definition AS+, Siemens). Patients were scanned in a head-first supine position. No contrast agent was administered.

Scan parameters in center 1 were: mean exposure 28.0 ± 9.3 mAs, collimation 80 × 0.625 mm, pitch 0.763, tube voltage 120 kV, matrix 512 x 512, slice thickness 2 mm, overlap 1 mm, mean CTDIvol 1.9 ± 0.6 mGy, mean DLP 80.8 ± 26.5 mGy*cm.

Scan parameters in center 2 were: mean exposure 116.3 ± 41.3 mAs, collimation 38.4 × 0.6 mm, pitch 1.2, tube voltage 100–140 kV, matrix 512 x 512, slice thickness 1 mm, overlap 0 mm, mean CTDIvol 7.6 ± 2.5 mGy, mean DLP 262.6 ± 90.2 mGy*cm.

## Calcium quantification

Quantification of AWC was performed using the vendor's software (IntelliSpace Portal, HeartBeat-CS, Philips Healthcare). The software highlighted all regions with a minimum volume of 0.5 mm$^3$ and attenuation above 130 HU with a color-coded overlay (Fig 1), as originally described by Janowitz, Agatston et al. [30]. For different tube voltages than 120 kV, the attenuation threshold was adjusted based on reported data [31]. The attenuation thresholds were 145.0 HU for 100 kV tube voltage, 136.0 HU for 110 kV, 126.1 HU for 130 kV and 123.4 HU for 140 kV. One radiologist (experience in chest CT imaging of more than two years) manually confirmed highlighted lesions with a mouse click. Aortal plaques were included from the ascending thoracic aorta to the diaphragm. Aortic valve calcifications and calcifications associated with the coeliac trunk were excluded from the measurements. The software automatically performed three-dimensional volumetry without specific user interaction of all connected voxels in each marked lesion above the attenuation threshold, after all, automatic segmentations were manually validated. The total of AWC in mm$^3$ was noted.

## Statistics and data analysis

Statistical analysis was performed in R language for statistical computing, R Foundation, Vienna, Austria, version 4.0.0. Multivariate regression was used to model the relationship between the dependent variable "severe COVID-19" and the regressors AWC as well as the control variables patient age, sex and cardiovascular comorbidity score. Magnitude of AWC in mm$^3$ was transformed to a logarithmic scale before multivariate regression to reduce impact of outliers. For patients without measurable AWC, a substitute AWC of 1 mm$^3$ was assumed. Multivariate regression was implemented by the glm() function in "binomial" mode. Visualization of the regression curve was achieved by the visreg package for R, version 2.7.0 [32]. To illustrate the accuracy of fit of the logistic regression, pseudo-$R^2$ was calculated as described by Nagelkerke ($> 0.2$ acceptable, $> 0.4$ good, $> 0.5$ very good) [33]. To preclude multicollinearity of the independent variables, the variance inflation factor (VIF) was calculated for each independent variable using the cars package for R ($< 2.5$ no significant multicollinearity) [34, 35].

Wilcoxon test was performed for comparison of ordinal data (outcome, clinical comorbidity score) and non-parametric data (AWC), chi-squared test for nominal data (patient sex), and two-sided T-test for parametric data (patient age).

All mentioned values are stated as average ± standard deviation, if not otherwise specified. Statistical significance was defined as $p \leq 0.05$.

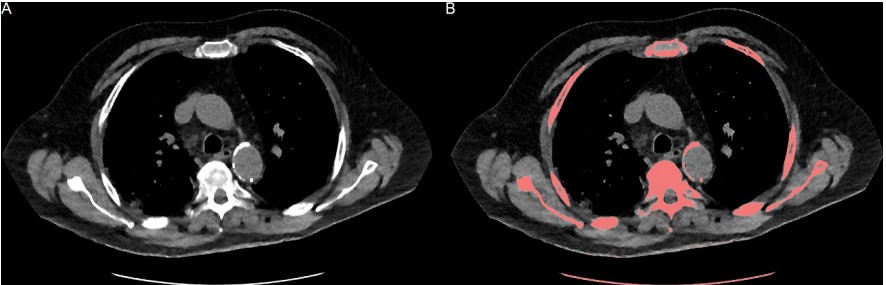

**Fig 1. Quantification of aortic wall calcification. A**: Low-dose chest CT (LDCT) slice in a soft tissue window at the height of the aortic arch. **B**: Connected voxels with an attenuation above the tube voltage specific threshold and a minimum volume of 0.5 mm$^3$ were highlighted by dedicated software. Plaques of the thoracic aorta were manually selected, and the sum of their volume was noted.

**Table 1. Patient details.**

|  | **Moderate COVID-19 (n = 38)** | **Severe COVID-19 (n = 32)** |
|---|---|---|
| Center (1/2) | 30/8 | 15/17 |
| Gender (m/f) | 21/17 | 20/12 |
| Patient age in years*[a] | 54.2 ± 14.1 | 67.0 ± 14.3 |
| Cardiovascular comorbidity score (0–5)*[b] | 1.3 ± 1.3 | 2.2 ± 1.2 |
| AWC in mm$^{3}$*[b] | median 0 (Q1 = 0, Q3 = 57.3) | median 771.7 (Q1 = 49.8, Q3 = 3065.5) |

Patient age, cardiovascular comorbidity score and aortic wall calcification (AWC) were significantly higher for individuals with severe coronavirus disease 2019 (COVID-19)

*[a] two-sided T-test

*[b] Wilcoxon test

$p \leq 0.05$. Male sex was not significantly higher throughout severe COVID-19 cases (Chi-square test, $p = 0.71$).

## Results

A total of 70 patients, 41 men and 29 women with a mean age of 60.0 ± 15.5 years were included; 45 patients admitted to center 1 and 25 patients to center 2 (Table 1). Of all patients included, 38 developed moderate COVID-19 (ambulant, hospitalization) and 32 a severe progression up to death (ICU, tracheal intubation, death).

Throughout the severe COVID-19 cases (n = 32), 20 patients were male. There was no significant predominance of male patients in the severe subgroup (p = 0.71). Patients with severe COVID-19 complications were significantly older and had a significantly higher burden of AWC than the moderate COVID-19 cases (patient age 67.0 ± 14.3 years vs. 54.2 ± 14.1 years, $p \leq 0.05$; AWC median 771.7 mm$^3$ (Q1 = 49.8 mm$^3$, Q3 = 3065.5 mm$^3$) vs. median 0 mm$^3$ (Q1 = 0 mm$^3$, Q3 = 57.3 mm$^3$), $p \leq 0.05$). Severe COVID-19 cases more frequently reported history of cardiovascular comorbidity (comorbidity score 1.3 ± 1.3 vs. 2.2 ± 1.2, $p \leq 0.05$).

Within multivariate logistic regression modeled by the independent variables of patient age, sex, cardiovascular comorbidity score and magnitude of AWC, the latter was the only significant regressor for severe COVID-19 (p = 0.004). The regression model notably benefitted from inclusion of AWC in addition to the demographic and clinical risk factors (Table 2).

For AWC > 3000 mm$^3$, average patient age, sex and cardiovascular comorbidity score, the regression predicted a risk of severe COVID-19 of 0.78 (95% CI 0.46–0.94). Without visually detectable AWC, the risk of severe COVID-19 was 0.13 (95% CI 0.04–0.36) (Fig 2). The regression's accuracy of fit is represented by a pseudo-$R^2$ = 0.43, as calculated by the Nagelkerke method. Multicollinearity among the independent variables is ruled out by a low VIF for each

**Table 2. Multivariate regression for prediction of severe COVID-19.**

| Consecutively included independent variables | Pseudo-$R^2$ (Nagelkerke) |
|---|---|
| Demographic data (patient age and sex) | 0.23 ("acceptable") |
| . . . clinical data (anamnestic comorbidity score) | 0.29 ("acceptable") |
| . . . CT-derived data (AWC) | 0.43 ("good") |

The accuracy of fit of our multivariate logistic regression model to predict severe COVID-19 benefitted from including clinical data in addition to generic demographic patient details. The best accuracy of fit was achieved by our final model including demographic, clinical and CT-derived data (Pseudo-$R^2$ 0.43).

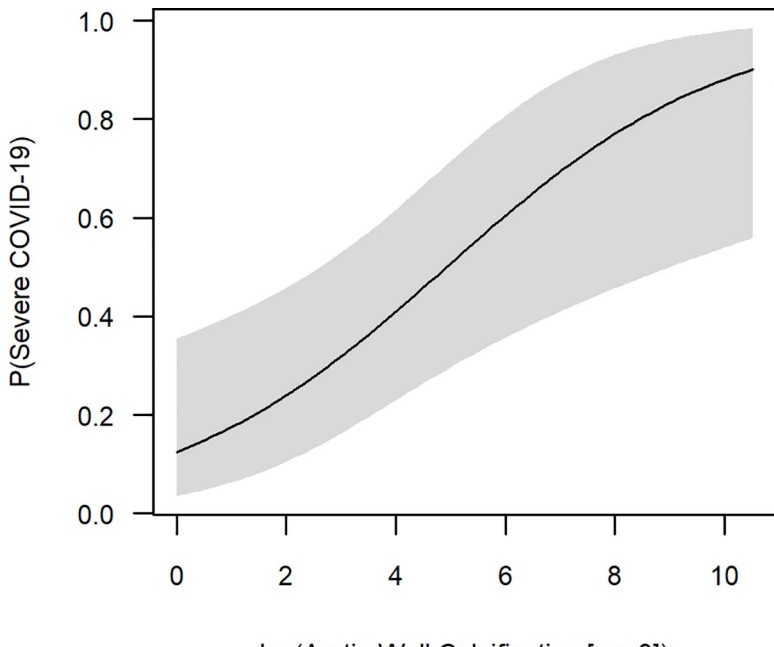

**Fig 2. Multivariate logistic regression for prediction of severe COVID-19.** Multivariate regression was modeled including the independent variables patient age, sex, an anamnestic cardiovascular comorbidity score and the magnitude of aortic wall calcification (AWC). Logistic regression was fit to a logarithmic scale for AWC to reduce the impact of outliers. Within our model, AWC was the only significant regressor to predict severe COVID-19 (p = 0.004). The 95% confidence interval is illustrated by a grey band (alpha = 0.05).

explanatory variable. The VIF was 1.9, 1.9, 1.0 and 1.1 for AWC, patient age, sex and cardiovascular comorbidity score, respectively.

## Discussion

From the very beginning of the pandemic, Chinese authors recognized a correlation of CRFs and cardiovascular comorbidities with a severe progression of the disease [2, 6, 7]. This observation was consistently confirmed by international authors [3, 21–25]. CRFs have further historically been linked to poor outcomes of infectious lung disease in general, such as an increase of myocardial infarctions during the influenza season [36]. Before the COVID-19 pandemic, magnitude of AWC has been shown to predict higher mortality of pneumonia without limitation to a specific infectious agent [37]. However, the biological causalities between CRFs and severe COVID-19 often still remain unexplained. Recent studies focus on the renin-angiotensin system since SARS-CoV-2 uses the angiotensin-converting enzyme 2 receptor (ACE2) for internalization to the host cell [22]. For individuals with chronic heart disease and diabetes, the host cell's altered receptor status might promote severe progression of COVID-19 [22]. Nicotine abuse increases the gene expression of ACE2 in lung tissue [24], which might facilitate disease progression in smokers. ACE inhibitors and angiotensin II type-I receptor blockers, which are commonly prescribed in the context of arterial hypertension, have also been suspected to increase the risk of developing severe and fatal COVID-19 [21, 23]. Disregarding the exact biological pathways CRFs and cardiovascular comorbidities are significantly associated with severe COVID-19 and can be assessed by quantification of atherosclerotic vessel calcification.

The short-term 5-day outcome of COVID-19 has been successfully predicted by quantification of lung involvement [38]. However, the extent of pulmonary changes most rapidly evolves

during the first week of the disease [39]. This renders the outcome predictor "extent of lung involvement" highly susceptible to the time point of imaging during early COVID-19. For best individual patient outcome an early assessment of whether hospitalization or intensive care treatment might be necessary is required. In our study, the amount of AWC was the only significant regressor for the long-term 22-day outcome of COVID-19. AWC derive from chronic CRFs and do not vary short-term. Our logistic regression model notably profited by the inclusion of AWC in addition to demographic and clinical data. Pseudo-$R^2$ by the Nagelkerke method was 0.43, which is rated a "good" accuracy of fit [33]. This implies that 43% of the variance for the probability of severe progression of COVID-19 can be explained by our model. In comparison, a regression model including patient age, sex and clinical comorbidity score only achieved a pseudo-$R^2$ of 0.29. Assuming average values for patient age, sex and clinical comorbidity score, our model predicts the obligation for intensive care treatment as 0.78 for AWC > 3000 mm$^3$. *Vice versa*, if there are no visually detectable AWC, the probability for admission to ICU is only 0.13.

AWC represent the patient's atherosclerotic burden, which results from lifestyle-related risk factors and demographic parameters such as patient age or sex [37]. The assessment of a patients' cardiovascular comorbidity by quantification of AWC as a surrogate might prove favorable in the context of the current pandemic. Calcium scoring of the thoracic aorta can be performed in a few seconds in daily routine CT reporting. Moreover, fully automized, deep learning based approaches to assess AWC have been investigated, and first clinical solutions are available [40–42]. In comparison to a physical examination and individual in-depth anamnesis, calcium scoring on LDCT for assessment of cardiovascular comorbidity does not deplete personal protection equipment and does not add to the infection risk for medical professionals. In our study, a combination of simple clinical data and scoring of AWC yielded the most accurate results (Table 2). The below threshold VIF of AWC (1.9) in our regression model confirms that AWC is not merely a recombination of the control variables patient age, sex and clinical comorbidity score, but an independent biomarker for upcoming severe complications of COVID-19.

Limitations of our study are the relatively small patient population and simplified endpoints of the statistical analysis. Comorbidities were simplified to easily obtainable items, which recognizes that a comprehensive clinical examination and anamnesis might not be achievable for every individual during the pandemic. Future studies should include a larger number of patients in a prospective multi-center, multi-vendor approach and focus on automated processing of the LDCT scans, possibly identifying further risk factors for severe COVID-19 progression.

## Conclusion

During the current COVID-19 pandemic hospitals are at risk to decompensate due to unprecedented numbers of patients. Physicians from regional COVID-19 hotspots reported overwhelming numbers of patients resulting in a deficiency of ICU beds [43–45]. When medical resources have to be rationalized, it is crucial to identify patients in need of intensive care and invasive ventilation as early as possible [44]. Our study demonstrates a correlation of a severe clinical manifestation of COVID-19 and a high atherosclerotic burden of the thoracic aorta, measured at the time of patient admission to the hospital, which is an average of 12 days before the start of intensive care treatment [2]. Further prospective multi-center, multi-vendor studies should investigate if opportunistic calcium scoring can anticipate the number of required ICU beds in advance, which might facilitate resource management and promote collaboration of hospitals on a national and international scale.

## Supporting information

**S1 Data. Patient details.**
(CSV)

**S1 Fig. Striking figure.** Soft tissue and lung window of an axial computed tomography are combined to illustrate the typical pulmonary findings of COVID-19 besides calcification of the thoracic aorta (light red).
(TIF)

## Author Contributions

**Conceptualization:** Philipp Fervers, Jonathan Kottlors, David Maintz, Stephanie Tritt, Nuran Abdullayev, Thorsten Persigehl.

**Data curation:** Philipp Fervers, Jonathan Kottlors, Orkhan Safarov, Nuran Abdullayev.

**Formal analysis:** David Zopfs, Johannes Bremm, David Maintz, Orkhan Safarov, Stephanie Tritt, Nuran Abdullayev, Thorsten Persigehl.

**Investigation:** Philipp Fervers, Jonathan Kottlors, David Zopfs, Johannes Bremm, Orkhan Safarov, Stephanie Tritt, Nuran Abdullayev, Thorsten Persigehl.

**Methodology:** Philipp Fervers, David Zopfs, Johannes Bremm.

**Supervision:** David Maintz, Thorsten Persigehl.

**Validation:** David Maintz, Thorsten Persigehl.

**Writing – original draft:** Philipp Fervers, Thorsten Persigehl.

**Writing – review & editing:** Philipp Fervers, Jonathan Kottlors, David Zopfs, Johannes Bremm, David Maintz, Orkhan Safarov, Stephanie Tritt, Nuran Abdullayev, Thorsten Persigehl.

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
