## [Decision Letter · Decision Letter 0]

1 Oct 2020

PONE-D-20-23872

Calcification of the thoracic aorta on low-dose chest CT predicts severe COVID-19

PLOS ONE

Dear Dr. Fervers,

Thank you for submitting your manuscript to PLOS ONE. After careful consideration, we feel that it has merit but does not fully meet PLOS ONE’s publication criteria as it currently stands. Therefore, we invite you to submit a revised version of the manuscript that addresses the points raised during the review process.

ACADEMIC EDITOR: All issues raised by expert reviewers are required.

We look forward to receiving your revised manuscript.

Kind regards,

Vincenzo Lionetti, M.D., PhD

Academic Editor

PLOS ONE

Journal Requirements:

2.Thank you for stating the following in the Competing Interests section:

[I have read the journal's policy and the authors of this manuscript have the following competing interests:

David Maintz has received speaker’s honoraria from Philips Healthcare, unrelated to the presented work.

David Zopfs has received exemption from clinical duties as part of a research agreement between Philips Healthcare and University Hospital Cologne, unrelated to this project.].

3. Please ensure that you refer to Figure 2 in your text as, if accepted, production will need this reference to link the reader to the figure.

Reviewers' comments:

Reviewer's Responses to Questions

**Comments to the Author**

1. Is the manuscript technically sound, and do the data support the conclusions?

Reviewer #1: Partly

Reviewer #2: Partly

2. Has the statistical analysis been performed appropriately and rigorously? 

Reviewer #1: Yes

Reviewer #2: No

3. Have the authors made all data underlying the findings in their manuscript fully available?

Reviewer #1: Yes

Reviewer #2: Yes

4. Is the manuscript presented in an intelligible fashion and written in standard English?

Reviewer #1: Yes

Reviewer #2: No

5. Review Comments to the Author

Reviewer #1: The Authors investigated aortic wall calcification on chest CT in 70 COVID-19 patients, as a surrogate of atherosclerotic burden, and its relationship with the severity of COVID-19 clinical manifestations. The topic is interesting, but the analysis needs to explore some points deeper.

-The results that "Multivariate regression analysis including AWC, patient age and sex, volume of AWC was the only significant regressor for severe COVID-19" indicated the the Authors did not take into account other clinical and demographic data. This has been acknowledged in the discussion: "Comorbidities of patients were not respected, since our aim was to model a prediction solely based on imaging and generic patient data. This recognizes that a comprehensive clinical examination and anamnesis might not be achievable for every individual during the pandemic". Actually, the collection of simple clinical data (BMI/obesity, history of hypertension, diabetes, hypercolesterolemia, smoking habit, prior myocardial infarction, drugs...) would be necessary to explore whether AWC predicts COVID-19 outcome independently from a simple anamnestic score, or whether AWC is a marker of the cardiovascular burden, useful only for patients whose anamnesis is not available. Most COVID-19 patients are indeed able to provide simple clinical data (medical history and drugs) to reconstruct their cardiovascular history and risk, so that the analysis of AWC might present very limited clinical usefulness.

"Outcome was rated moderate (no hospitalization, hospitalization) and severe (ICU, tracheal intubation, death)". It would be interesting to explore a more detailed outcome score (1-5) and its relationship with AWC, rather then using a binary outcome (moderate 1-2 vs severe 3-4-5).

In the Methods, "Aortic valve calcifications and calcifications associated with the coeliac trunk were excluded." Why not considering calcium burden in other sites, such as aortic valve, abdominal vessels...? It might further stratify patients' risk.

In the Methods "All mentioned values are stated as average ± standard deviation, if not otherwise specified." Actually, AWC (Table 1) should be expressed as median (interquartile range), because it is clearly not normally distributed.

In the discussion, the Authors correctly stated that "The short-term 5-day outcome of COVID-19 has been successfully predicted by quantification of lung involvement" despite being "highly susceptible to the time point of imaging during early COVID-19." Nevertheless, I would try to explore whether the extent of lung involvement is somehow correlated to AWC and whether it provides additive prognostic information on top of AWC.

Reviewer #2: General comment:

Generally, this is an interesting paper, which seems to have significantly benefited from previous reviewers’ comments. The idea of the paper to quantify aortic calcium burden to help predicting the course of COVID19 might be a helpful tool in clinical practice. However, the authors may not over exaggerate the findings of a small retrospective study by including prediction assumptions but rather state the correlation and, potentially, recommend a prospective study setting.

Specific comments:

Introduction:

It is still fairly long and may benefit from further shortening.

The objective may be rewritten because AWC does not influence the course of COVID19 but might represent some kind of biomarker, which helps to predict the course of COVID 19.

Material and Methods:

LDCT scanning protocols: it seems the tube voltage of the CT scanner was 100kVp while the voltage of center 1 was 120 kVp. It is known that lowering the tube voltage increases the CT numbers of calcification, yielding false high measurements. Has this been compensated? Was the Philips software feasible to correct for this? Otherwise, measurements might be skewed.

It appears that the radiation dose of both centers differed significantly, and you analyzed images of different slice thicknesses, which is also prone to systematic errors. Has this problem been considered?

“For independence of imaging results” might be omitted, since it has not been tested.

Statistical and data analysis:

Multiple regression analyses should be screened for interactions of variables. Based on experience and literature, interactions of age, gender, and AWC are highly suspected.

Further demographic data (e.g., patient size) may be included in multiple regression analysis, as they will affect the total amount of AWC.

Results:

Discussion:

May be shortened and focused on the discussion of the study’s results. Repetitions from the introduction may be removed.

Discussion of cut-off values (L238-241) is not supported by the results section. These findings may be added to the results section.

Typos and punctation errors:

L19: The systemic

L22: the diagnosis

L25: Seventy

L26: The outcome

L28: a requirement

L28: The amount

L32: , and

L32: The average

L35: , including

L35: the volume

L37: predicts risk

L37: a severe

L48: i.e., fever

L53: The diagnosis

L55: transcription-polymerase

L57: beds in clinical

L57: practice, this

L60: the specificity

L62: false-negative

L62: eight

L64: , and diagnosis

L66: a history

L71: The evaluation

L73: , considering

L74: , more specifically,

L75: an atherosclerotic

L77: , and

L78: indicator of

L86: , and

L87: COVID-19 outcome and

L87: particular,

L94: and (no or)

L104: the 10th

L104: the acquisition

L107: is not in

L111: hospital-based

L111: on reported

L116: a more

L125: the independence

L140: two

L145: after all,

L149: in a soft

L150: a minimum

L150: by dedicated

L158: , which

L187: , and

L191: Multivariate regression ( no a )

L192: , including

L192: , and

L194: the impact

L200: change as to like

L200/201: time-consuming

L202: while

L208: , and

L212: outcomes

L213: the influenza

L217: angiotensin-converting

L219: the host cell's altered receptor status might promote severe

progression of COVID-19

L221: , which

L222: , which

L223: hypertension,

L232: study, the

L232: the long-term

L234: the inclusion

L235: , which

L236: the probability

L240: the probability

L244: lifestyle-related

L245: , and

L245: the development

L247: , which

L249: lifestyle-related

L250: the magnitude

L264: rationalized,

L267: , which

L267: the start

6. PLOS authors have the option to publish the peer review history of their article (what does this mean?). If published, this will include your full peer review and any attached files.

Reviewer #1: No

Reviewer #2: No

---

## [Author Response · Author response to Decision Letter 0]

18 Nov 2020

Response to the reviewer comments (see uploaded file "Response to Reviewers.docx"):

Reviewer #1: The Authors investigated aortic wall calcification on chest CT in 70 COVID-19 patients, as a surrogate of atherosclerotic burden, and its relationship with the severity of COVID-19 clinical manifestations. The topic is interesting, but the analysis needs to explore some points deeper.

-The results that "Multivariate regression analysis including AWC, patient age and sex, volume of AWC was the only significant regressor for severe COVID-19" indicated the the Authors did not take into account other clinical and demographic data. This has been acknowledged in the discussion: "Comorbidities of patients were not respected, since our aim was to model a prediction solely based on imaging and generic patient data. This recognizes that a comprehensive clinical examination and anamnesis might not be achievable for every individual during the pandemic". Actually, the collection of simple clinical data (BMI/obesity, history of hypertension, diabetes, hypercolesterolemia, smoking habit, prior myocardial infarction, drugs...) would be necessary to explore whether AWC predicts COVID-19 outcome independently from a simple anamnestic score, or whether AWC is a marker of the cardiovascular burden, useful only for patients whose anamnesis is not available. Most COVID-19 patients are indeed able to provide simple clinical data (medical history and drugs) to reconstruct their cardiovascular history and risk, so that the analysis of AWC might present very limited clinical usefulness.

Thank you very much for this very important comment which substantially improved our study. As suggested, we included an anamnestic risk score of cardiovascular comorbidity to our multivariate logistic regression. Our score considers relevant basic clinical data, which can be investigated in a quick interview and by reviewing a patient’s daily medication: 1. history of arterial hypertension or antihypertensive drugs in medication, 2. type 2 diabetes or anti-diabetic medication, 3. hyperlipidemia or lipid-lowering agents in medication, 4. active smoking, 5. prior cardiovascular event (myocardial infarction or revascularization procedure, stroke or transient ischemic attack, diagnosis of peripheral arterial disease). Each of the five risk factors was binarily scored as 0/1 and the sum was considered as a control variable in the multivariate logistic regression. By inclusion of the anamnestic risk score, the accuracy of our regression was increased from pseudo R²=0.41 to pseudo R²=0.43 (Nagelkerke).

"Outcome was rated moderate (no hospitalization, hospitalization) and severe (ICU, tracheal intubation, death)". It would be interesting to explore a more detailed outcome score (1-5) and its relationship with AWC, rather then using a binary outcome (moderate 1-2 vs severe 3-4-5).

A more detailed outcome on an ordinal scale could be modeled by an ordinal multivariate regression, but based on our “proof of principle” data this statement seems us to be too uncertain and was beyond the scope of this study. From our point of view this requires a much larger patient population and is planned to be investigated in an upcoming further larger multi-center, multi-vendor study.

In the Methods, "Aortic valve calcifications and calcifications associated with the coeliac trunk were excluded." Why not considering calcium burden in other sites, such as aortic valve, abdominal vessels...? It might further stratify patients' risk.

As you suggested, we additionally measured the calcium burden of the coeliac trunk and aortic valve (see S1_data). 

Since the coeliac trunk is located on the most caudal slides of the CT scan, it was frequently not entirely included in the scan’s volume of interest. To maintain consistency throughout our patient population, we refrained from including calcification at the coeliac trunk to our regression. 

Calcification of the aortic valve was a significant regressor for severe COVID-19 throughout univariate logistic regression (p<0.05). However, after including the control variables (patient age, sex, anamnestic risk score), calcification of the aortic valve did not significantly influence the probability for severe COVID-19 (p=0.17). In a multivariate regression model including aortic valve calcification, aortic wall calcification and the control variables, aortic wall calcification was the only significant regressor (p<0.05). Thus, we did not add calcification of the aortic valve to our final regression model.

In the Methods "All mentioned values are stated as average ± standard deviation, if not otherwise specified." Actually, AWC (Table 1) should be expressed as median (interquartile range), because it is clearly not normally distributed.

We agree. AWC is now expressed as median (interquartile range).

In the discussion, the Authors correctly stated that "The short-term 5-day outcome of COVID-19 has been successfully predicted by quantification of lung involvement" despite being "highly susceptible to the time point of imaging during early COVID-19." Nevertheless, I would try to explore whether the extent of lung involvement is somehow correlated to AWC and whether it provides additive prognostic information on top of AWC.

The aim of this study was to explore if atherosclerosis, as a surrogate of cardiovascular comorbidity, affects the long-term outcome of COVID-19. Our data are not uniform concerning the time period from first symptoms of COVID-19 to CT imaging. During the early phase of COVID-19, lung involvement rapidly evolves. I.e. between day 1 and day 2 of COVID-19, lung involvement score by Shuchang Zhou et al. approximately doubles (1). This relationship of time point and lung involvement in COVID-19 would fundamentally bias such analysis throughout our data. However, correlation of atherosclerosis with the extent of lung involvement might be a promising subject for further studies with defined time period from first symptoms of COVID-19 to CT imaging.

Reviewer #2: General comment:

Generally, this is an interesting paper, which seems to have significantly benefited from previous reviewers’ comments. The idea of the paper to quantify aortic calcium burden to help predicting the course of COVID19 might be a helpful tool in clinical practice. However, the authors may not over exaggerate the findings of a small retrospective study by including prediction assumptions but rather state the correlation and, potentially, recommend a prospective study setting.

The conclusions are toned down as you suggested. A larger prospective multi-center, multi-vendor study design is now recommended to determine the predictive power of aortic wall calcifications towards the outcome of COVID-19.

Specific comments:

Introduction:

It is still fairly long and may benefit from further shortening.

The objective may be rewritten because AWC does not influence the course of COVID19 but might represent some kind of biomarker, which helps to predict the course of COVID 19.

The introduction is shortened and the objective rewritten as you suggested.

Material and Methods:

LDCT scanning protocols: it seems the tube voltage of the CT scanner was 100kVp while the voltage of center 1 was 120 kVp. It is known that lowering the tube voltage increases the CT numbers of calcification, yielding false high measurements. Has this been compensated? Was the Philips software feasible to correct for this? Otherwise, measurements might be skewed.

It appears that the radiation dose of both centers differed significantly, and you analyzed images of different slice thicknesses, which is also prone to systematic errors. Has this problem been considered?

Thank you very much for this comment. After further literature research we agree that the issue of varying tube voltages in center 2 might skew our data. Several studies about quantification of coronary artery calcification state that with lower tube voltage, the magnitude of calcification is generally overestimated. To cope with this bias, an adjustment of the threshold density for identification of a vessel plaque has been suggested. The standard threshold by Agatston being 130 HU at 120 kV tube voltage, two studies identified 145 HU and 147 HU at 100 kV as the optimum threshold for best comparability to the standard method (2,3). The CT scans in our study from center 2 were acquired with tube voltage 100 kV – 140 kV, which was corrected in the materials and methods section. In order to identify the best threshold values for identification of vessel plaques, we fit an almost perfectly accurate exponential regression to the data by Grän et al. (r²=0.999, Fig 1).

Fig 1: The density threshold for identification of vessel plaques depends on the tube voltage. Grän et al. identified 4 optimal attenuation thresholds for different tube voltages in an experimental and mathematical fashion. We fit an exponential regression to their data (r²=0.999) and calculated the thresholds for 110 kV (136.0 HU), 130 kV (126.1 HU) and 140 kV (123.4 HU). 

Fabrice et al. argue that modification of the threshold alone cannot compensate for the difference in tube voltage when measuring coronary artery calcification (4). However, in our study we didn’t quantify coronary artery calcification, but aortal calcification, which consists of larger confluent plaques. Discrepancy in measuring calcified plaques arises on the outline of the calcification, which is often blurred and prone to blooming artifacts. Larger aortic plaques have a smaller surface-to-volume ratio and thus should be less affected by different tube voltages.

To compensate for the usage of different tube voltages, all measurements of calcification for patients from center 2 with different tube voltage than 120 kV were repeated with the adjusted thresholds. The multivariate regression model for prediction of severe COVID-19 improved by repeating the measurements with tube voltage adjusted thresholds. Pseudo R² for the new model (independent variables patient age, sex and aortic wall calcification) increased to 0.41 (Nagelkerke, “good accuracy”).

Modification of the tube current and consequently the radiation dose of a CT scan affects the standard deviation of measurements (noise) but not the average CT density (4). The different radiation doses in center 1 and 2 accordingly should not bias the measurements. Christensen et al. investigated the influence of slice thickness on the quantification of vessel plaques on axial CT scans and found that 1.25 mm and 2.5 mm slice thickness yield comparable results (5). Hence, in our study the difference between 1 mm slice thickness in center 2 and 2 mm in center 1 should not bias the measurements.

“For independence of imaging results” might be omitted, since it has not been tested.

The statement is omitted.

Statistical and data analysis:

Multiple regression analyses should be screened for interactions of variables. Based on experience and literature, interactions of age, gender, and AWC are highly suspected.

Variance inflation factor (VIF) was calculated for each independent variable of the logistic regression to screen for multicollinearity. 

Further demographic data (e.g., patient size) may be included in multiple regression analysis, as they will affect the total amount of AWC.

A cardiovascular risk score with further anamnestic data was added to our multivariate logistic regression model as explained above (1. history of arterial hypertension or antihypertensive drugs in medication, 2. type 2 diabetes or anti-diabetic medication, 3. hyperlipidemia or lipid-lowering agents in medication, 4. active smoking, 5. prior cardiovascular event). Patient size might be another interesting parameter to add as a control variable in a prospective study design. However, retrospectively we do not have sufficient data in the medical records to support such analysis. Patient size might also be estimated based on the CT-scans. Yet, this would not serve the purpose of a non-CT-based control variable.

Results:

Discussion:

May be shortened and focused on the discussion of the study’s results. Repetitions from the introduction may be removed.

The discussion is shortened. In particular repetitions from the introduction are removed.

Discussion of cut-off values (L238-241) is not supported by the results section. These findings may be added to the results section.

Cut-off values are added to the results section.

Typos and punctation errors:

L19: The systemic

L22: the diagnosis

L25: Seventy

L26: The outcome

L28: a requirement

L28: The amount

L32: , and

L32: The average

L35: , including

L35: the volume

L37: predicts risk

L37: a severe

L48: i.e., fever

L53: The diagnosis

L55: transcription-polymerase

L57: beds in clinical

L57: practice, this

L60: the specificity

L62: false-negative

L62: eight

L64: , and diagnosis

L66: a history

L71: The evaluation

L73: , considering

L74: , more specifically,

L75: an atherosclerotic

L77: , and

L78: indicator of

L86: , and

L87: COVID-19 outcome and

L87: particular,

L94: and (no or)

L104: the 10th

L104: the acquisition

L107: is not in

L111: hospital-based

L111: on reported

L116: a more

L125: the independence

L140: two

L145: after all,

L149: in a soft

L150: a minimum

L150: by dedicated

L158: , which

L187: , and

L191: Multivariate regression ( no a )

L192: , including

L192: , and

L194: the impact

L200: change as to like

L200/201: time-consuming

L202: while

L208: , and

L212: outcomes

L213: the influenza

L217: angiotensin-converting

L219: the host cell's altered receptor status might promote severe

progression of COVID-19

L221: , which

L222: , which

L223: hypertension,

L232: study, the

L232: the long-term

L234: the inclusion

L235: , which

L236: the probability

L240: the probability

L244: lifestyle-related

L245: , and

L245: the development

L247: , which

L249: lifestyle-related

L250: the magnitude

L264: rationalized,

L267: , which

L267: the start

Typos and punctuation errors are corrected as suggested.

References

1. Zhou S, Zhu T, Wang Y, Xia LM. Imaging features and evolution on CT in 100 COVID-19 pneumonia patients in Wuhan, China. Eur Radiol [Internet]. 2020 May 4 [cited 2020 Jul 30];1–9. Available from: https://doi.org/10.1007/s00330-020-06879-6

2. Marwan M, Mettin C, Pflederer T, Seltmann M, Schuhbäck A, Muschiol G, et al. Very low-dose coronary artery calcium scanning with high-pitch spiral acquisition mode: Comparison between 120-kV and 100-kV tube voltage protocols. J Cardiovasc Comput Tomogr [Internet]. 2013 Jan [cited 2020 Oct 15];7(1):32–8. Available from: https://pubmed.ncbi.nlm.nih.gov/23333186/

3. Gräni C, Vontobel J, Benz DC, Bacanovic S, Giannopoulos AA, Messerli M, et al. Ultra-low-dose coronary artery calcium scoring using novel scoring thresholds for low tube voltage protocols—a pilot study. Eur Heart J Cardiovasc Imaging [Internet]. 2018 [cited 2020 Oct 15];19(12):1362–71. Available from: https://pubmed.ncbi.nlm.nih.gov/29432592/

4. Deprez FC, Vlassenbroek A, Ghaye B, Raaijmakers R, Coche E. Controversies about effects of low-kilovoltage MDCT acquisition on Agatston calcium scoring. J Cardiovasc Comput Tomogr. 2013 Jan 1;7(1):58–61. 

5. Christensen JL, Sharma E, Gorvitovskaia AY, Watts JP, Assali M, Neverson J, et al. Impact of slice thickness on the predictive value of lung cancer screening computed tomography in the evaluation of coronary artery calcification. J Am Heart Assoc [Internet]. 2019 Jan 1 [cited 2020 Oct 15];8(1). Available from: https://www.ahajournals.org/doi/10.1161/JAHA.118.010110

---

## [Decision Letter · Decision Letter 1]

8 Dec 2020

Calcification of the thoracic aorta on low-dose chest CT predicts severe COVID-19

PONE-D-20-23872R1

Dear Dr. Fervers,

We’re pleased to inform you that your manuscript has been judged scientifically suitable for publication and will be formally accepted for publication once it meets all outstanding technical requirements.

Kind regards,

Vincenzo Lionetti, M.D., PhD

Academic Editor

PLOS ONE

Additional Editor Comments (optional):

Reviewers' comments:

Reviewer's Responses to Questions

**Comments to the Author**

1. If the authors have adequately addressed your comments raised in a previous round of review and you feel that this manuscript is now acceptable for publication, you may indicate that here to bypass the “Comments to the Author” section, enter your conflict of interest statement in the “Confidential to Editor” section, and submit your "Accept" recommendation.

Reviewer #1: All comments have been addressed

Reviewer #2: All comments have been addressed

2. Is the manuscript technically sound, and do the data support the conclusions?

Reviewer #1: Yes

Reviewer #2: Yes

3. Has the statistical analysis been performed appropriately and rigorously? 

Reviewer #1: Yes

Reviewer #2: Yes

4. Have the authors made all data underlying the findings in their manuscript fully available?

Reviewer #1: Yes

Reviewer #2: Yes

5. Is the manuscript presented in an intelligible fashion and written in standard English?

Reviewer #1: Yes

Reviewer #2: Yes

6. Review Comments to the Author

Reviewer #1: (No Response)

Reviewer #2: The manuscript has been revised thoroghly. The results presented are of substantical interest to the readership of plos one. The paper puts forward a sound methodology. Methodical issues have been addressed and solved adequately in the revision.

7. PLOS authors have the option to publish the peer review history of their article (what does this mean?). If published, this will include your full peer review and any attached files.

Reviewer #1: No

Reviewer #2: **Yes: **Christoph Artzner

---

## [Editor Report · Acceptance letter]

14 Dec 2020

PONE-D-20-23872R1 

Calcification of the thoracic aorta on low-dose chest CT predicts severe COVID-19 

Dear Dr. Fervers:

I'm pleased to inform you that your manuscript has been deemed suitable for publication in PLOS ONE. Congratulations! Your manuscript is now with our production department. 

Kind regards, 

on behalf of

Prof. Vincenzo Lionetti 

Academic Editor

PLOS ONE